# Acute high folic acid treatment in SH-SY5Y cells with and without *MTHFR* function leads to gene expression changes in epigenetic modifying enzymes, changes in epigenetic marks, and changes in dendritic spine densities

**Daniel F. Clark, Rachael Schmelz, Nicole Rogers, Nuri E. Smith**[ID]¤**, Kimberly R. Shorter**[ID]*

Division of Natural Sciences and Engineering, University of South Carolina Upstate, Spartanburg, South Carolina, United States of America

¤ Current address: Lieber Institute for Brain Development, Johns Hopkins School of Medicine, Baltimore, Maryland, United States of America
* shortekr@uscupstate.edu

## Abstract

Epigenetics are known to be involved in various disorders, including neurobiological disorders like autism. Dietary factors such as folic acid can affect epigenetic marks using methylenetetrahydrofolate reductase (*MTHFR*) to metabolize folic acid to a one-carbon methyl group. As *MTHFR* mutations are frequent, it is curious as to whether excess folic acid, with or without functioning *MTHFR*, could affect gene expression, epigenetics, and neuromorphology. Here, we investigated gene expression and activity of epigenetic modifying enzymes, genome-wide DNA methylation, histone 3 modifications, and dendritic spine densities in SH-SY5Y cells with or without a knockdown of *MTHFR* and with or without an excess of folic acid. We found alterations to gene expression of epigenetic modifying enzymes, including those associated with disorders like autism. Grouping the epigenetic modifying enzymes by function indicated that gene expression was widely affected for genes that code for enzymes affecting DNA methylation, histone acetylation, histone methylation, histone phosphorylation, and histone ubiquitination when excess folic acid treatment occurred with or without the knockdown of *MTHFR*. *MTHFR* was significantly reduced upon excess folic acid treatment whether *MTHFR* was knocked-down or not. Further, methyl-CpG binding protein 2 expression was significantly decreased with excess folic acid treatment with and without proper *MTHFR* expression. Global DNA methylation decreased due to the knockdown alone while global hydroxymethylated DNA increased due to the knockdown alone. *TET2* expression significantly increased with the *MTHFR* knockdown alone. Excess folic acid alone induced a decrease in *TET3* expression. Excess folic acid induced an increase in dendritic spines without the *MTHFR* knockdown, but folic acid induced a decrease in dendritic spines when *MTHFR* was knockdown. The knockdown alone also increased the dendritic spines significantly. Histone 3

**Data Availability Statement:** All relevant data are within the manuscript and its Supporting information files.

**Funding:** Magellan Scholar Grant (Undergraduate Research Office/Vice President for Research at University of South Carolina; https://www.sc.edu/about/offices_and_divisions/undergraduate_research/index.php) to R. S. Magellan Mentor Grant (Office of Sponsored Awards and Research Support at the University of South Carolina Upstate; https://www.uscupstate.edu/research/sponsored-awards-and-research-support/contact-sars/) to K. S. Research Assistantship (Office of Sponsored Awards and Research Support at the University of South Carolina Upstate; https://www.uscupstate.edu/research/sponsored-awards-and-research-support/contact-sars/) to D. C. and K. S. Lab funds in the Natural Sciences and Engineering Division at the University of South Carolina Upstate (https://www.uscupstate.edu/) to K.S.

**Competing interests:** No authors have competing interests.

acetylation at lysine 18 was significantly increased when excess folic acid was applied to cells with the *MTHFR* knockdown, as was histone 3 phosphorylation at serine 10. Broadly, our results indicate that excess folic acid, even with functioning *MTHFR*, could have detrimental effects on cells.

## Introduction

Epigenetic mechanisms, including DNA methylation and histone modifications (acetylation, methylation) affect gene expression. Aberrant epigenetic marks are well understood to be part of the etiology of many diseases including heart disease, cancers, and psychiatric illnesses [1, 2]. Epigenetic modifications are affected by stress and dietary factors. Folic acid (FA) is a dietary supplement that affects epigenetic marks. FA is metabolized to a one carbon methyl group that can be added to DNA and/or histones. During FA metabolism, the methylenetetrahydrofolate reductase (MTHFR) enzyme is necessary.

FA supplementation in grain products began in the 1990s to prevent neural tube defects [3, 4]. FA is now widely consumed in vitamins/supplements and in fortified grain products [4]. Excess FA is not simply excreted from the body; FA fortification leads to higher blood FA concentrations in exposed persons [5]. Previous research on high FA consumption in rodent models has indicated excess FA consumption may induce detrimental physiological effects, including but not limited to behavioral/neurological changes and cataract formation [6–11]. Some studies indicate excess FA exposure could have detrimental effects in humans while others offer conflicting evidence [12–14].

Further complicating the FA dosage issue is an estimated 60% of the United States population has at least one copy of a mutation in the methylenetetrahydrofolate reductase (*MTHFR*) gene [4]. The *MTHFR* gene codes for an enzyme, methylenetetrahydrofolate reductase, which is, in part, responsible for FA metabolism in the one-carbon metabolic pathway. The mutation leads to a non-functioning *MTHFR* enzyme. Women with an *MTHFR* mutation are encouraged to supplement their diet with up to 10x the FDA's recommended daily amount of FA [4]. Prior studies have shown high FA supplementation leads to a significant decrease in MTHFR enzyme function in hepatocytes [15]. Since a heterozygous woman may or may not pass the *MTHFR* mutation to their offspring, it is possible that over supplementing FA, especially during pregnancy, could have detrimental effects in the fetus, as the fetus may or may not have inherited a mutant form of *MTHFR*. If the offspring has wild-type copies for both copies of *MTHFR*, excess FA may still have detrimental effects as seen in prior studies. But if the offspring does inherit at least one mutant copy of *MTHFR*, excess FA may lead to further decreased MTHFR enzyme function, which could be of further detriment to the developing fetus.

It is possible that an MTHFR enzyme malfunction—due to the mutation and/or FA overconsumption—could induce a lack of proper FA metabolism and a subsequent change to epigenetic marks such as DNA and histone methylation. Therefore, we conducted an exploratory study to determine the consequences of a 10x FA exposure with and without a knockdown of *MTHFR* on epigenetic marks, gene expression, and cell biology of SHSY5Y cells. SHSY5Y cells are a human neuroblastoma cell line that is commonly utilized as a neurobiological model [16–20]. DNA methylation, DNA hydroxymethylation, and histone modifications to histone 3 (H3) were assayed. Gene expression levels of epigenetic chromatin modifying enzymes and of *MTHFR* were also measured. Enzymatic activity of HATs (histone acetyltransferases), HDACs

(histone deacetylases), H3K4 HMTs (histone methyltransferases), TETs (ten-eleven transloca-
tion methylcytosine dioxygenases), and DNMTs (DNA methyltransferases) were examined as
well. Finally, dendritic spine densities were analyzed as a marker of cellular changes in our cell
line. It was hypothesized that a high FA treatment would affect gene expression of epigenetic
modifying enzymes, H3 modifications, global DNA methylation and hydroxymethylation, and
dendritic spine densities in the SHSY5Y cell line since *MTHFR* is necessary for FA metabo-
lism, and high FA treatments are known to further decrease *MTHFR* function.

## Materials and methods

### Cell culture and treatments

No institutional ethics approval was required for this study.

SHSY5Y cells were obtained from ATCC.org (Cat. No. CRL-2266 at www.ATCC.org) and
were grown in T75 tissue culture treated flasks with full growth medium and were kept at
37˚C with 5% $CO_2$ and high humidity. SHSY5Y cells are commonly utilized in neurobiological
studies [16–20]. The full growth medium was DMEM:F12 (Fisher Scientific, Cat. No.
MT10090CV) supplemented with 10% fetal bovine serum (FBS; Fisher Scientific; Cat. No.
MT35010CV) and 1% penicillin/streptomycin (Fisher Scientific; Cat. No. 15-140-148). Cells
were passaged at 80% confluence with 0.05% trypsin-EDTA (Fisher Scientific; Cat. No. 25-
300-054) and were plated in 6-well tissue culture treated plates (Fisher Scientific; Cat. No. 07-
200-83) at 200,000 cells/well. Cells were not used for experimentation if they were beyond pas-
sage 6.

RNAi was performed 24 hours after cells were plated. RNAi procedures consisted of treat-
ment with either a scramble siRNA (control) or an siRNA to *MTHFR* (knockdown). All treat-
ment groups received Lipofectamine RNAiMax Reagent (LifeTechnologies; 13-778-100) to
improve transfection efficiency. After the 48 hour RNAi period, cells were treated with either
water (control) or a 10x FA treatment (folic acid; Sigma; Cat. No. F8758-5G). The treatment
groups were as follows: (1) scramble siRNA+water, hereafter referred to as SCR, (2) scramble
siRNA+10x FA, hereafter referred to as SCR+10x FA, (3) *MTHFR* siRNA+water, hereafter
referred to as KD, and (4) *MTHFR* siRNA+10x FA, hereafter referred to as KD+10x FA (siR-
NAs from ThermoFisher; Cat. No's: Negative Control: AM4611, *MTHFR* siRNA: AM16708).
The DMEM:F12 medium contains 2.65mg/L FA. The 10x FA treatment groups were exposed
to a 26.5mg/L FA concentration. FA treatments lasted 48 hours before cells were harvested for
further applications. There were three replicates per treatment group. The contents of each
replicate are described below in each experimental procedure's section. Experiments were
repeated independently on different days. Experimenters were unaware of group assignments
during all of the following methods.

### Statistical methods

For all data sets, Excel was used to perform statistical analysis via a two-way ANOVA with rep-
lication. This was done to determine whether the KD alone, FA treatment alone, or an interac-
tion between the two independent variables was significant. An alpha level of 5% was used to
determine if there was significance.

### DNA isolation and DNA methylation & hydroxymethylation analysis

DNA was isolated using PureLink Genomic DNA Mini Kit (ThermoFisher; Cat. No.
K182000). There were three replicates per treatment group for the 5-mC kit and four replicates
per treatment group for the 5-hmC kit. Each replicate contained the contents of two wells, and

each well was from a different 6-well plate. The experiments were repeated in different months. The three replicates per treatment group were pooled and analyzed in duplicate on 96-well plates in ELISA-based assays for global DNA methylation (5-mC) (Epigentek kit; Cat. No. P-1030-96) and four samples were assayed in duplicate for global DNA hydroxymethylation (5-hmC) (Epigentek kit; Cat. No. P-1032-48).

For the 5-mC kit, pooled DNA samples and four DNA standards were incubated (in duplicate) with antibody for 5-methyl Cytosine (5-mC) for one hour. Washes followed and a secondary antibody with a conjugated reporter was then added. The wells were washed again and a substrate was added followed by a solution to inhibit the reporter-substrate reaction. The wells were analyzed for absorbance at a 450nm wavelength using a spectrophotometric plate reader. The percentage of 5-mC was determined for each sample. All calculations were completed according to instructions provided in the manual included in the Epigentek kit.

For the 5-hmC kit, each of 4 DNA samples per treatment group and seven DNA standards (in duplicate) were incubated with an antibody for 5-hydroxymethyl Cytosine (5-hmC) for one hour. A very similar protocol to the 5-mC kit followed; the wells were washed and a secondary antibody with a conjugated reporter was then added. The wells were washed again and a substrate was added followed by a solution to inhibit the reporter-substrate reaction. The wells were analyzed for absorbance at a 450nm wavelength using a spectrophotometric plate reader. The percentage of 5-hmC was calculated for each sample according to instructions provided in the manual included with the Epigentek kit.

## Histone isolations and histone modification analysis

Histones were isolated using a Histone Extraction Kit (Epigentek; Cat. No. OP-0007-192) and were quantified by Bradford assay. Three replicates per treatment group were obtained; each replicate contained the contents of 12 wells from two separate 6-well plates. Equal amounts of histone were then loaded into Histone 3 (H3) Multiplex Modification Array Kits (Epigentek; Cat. No. P-3100-96). Each plate contained 4 wells with antibodies for a specific histone modification to H3. One well was used for each sample type. A total of 6 array plates were used in order to test the 3 replicates in duplicate. After incubation with the antibodies, wells were washed and a secondary antibody with a conjugated reporter was incubated with the samples. A substrate was added followed by a solution to stop the reporter-substrate reaction and the wells were analyzed at a 450nm wavelength with a spectrophotometric plate reader. Analyses were performed according to the manual provided by Epigentek. The percentage of each modification was calculated as a function of the H3 modification level compared to the total H3 present in the sample.

## Cell staining and dendritic spine analysis

Cells were plated at 15,000 cells/coverslip on poly-D-lysine coated glass coverslips (Fisher Scientific; Cat. No. NC0343705). There were three coverslips per treatment group in separate dishes. Experiments for dendritic spine analysis were repeated during a different month. RNAi and FA treatments were performed as previously described. Afterwards, cells were stained with 1,1'-Dioctadecyl-3,3,3',3'-Tetramethylindocarbocyanine Perchlorate ("DiI" stain; Fisher Scientific; Cat. No. D3911) dissolved in DMSO (Fisher Scientific; Cat. No. MT25950CQC) as was done previously [21, 22].

Cell culture media was removed, and cells were fixed with 2% paraformaldehyde in PBS for 15 minutes. Cells were washed three times with Dulbecco's PBS (DPBS; Fisher Scientific; Cat. No. 14-190-144). Cells were then incubated in the dark at 37°C for 2 hours with warm DMEM:F12 media with 5uL/mL DiI stain solution and 4uL/mL Lipofectamine reagent. Cells were then washed thrice with warm DMEM:F12 for 30 minutes per wash. Cells were incubated

overnight in DPBS at room temperature in the dark. Counterstaining of nuclei was accomplished with DAPI (Fisher Scientific; Cat. No. D1306) at 5uL/mL in DPBS for 5 minutes at room temperature. Coverslips were mounted on slides with prolong gold antifade (Fisher Scientific; Cat. No. P36930) and were stored at 4˚C for two weeks before visualization with the confocal microscope. Images were analyzed using ImageJ software (NIH). Dendritic spine density was determined as number of spines per 10μm segment.

### RNA isolation and gene expression analysis

RNA was isolated using the Purelink RNA Isolation Kit (ThermoFisher; Cat. No. 12183018A). Three replicates per treatment group were obtained. Each replicate contained the contents of two wells from two separate 6-well plates. Experiments were repeated once during a different month. RNA was quantified using a nanodrop. RT-PCR reactions were setup using the High Capacity cDNA Synthesis Kit (ThermoFisher; Cat. No. 4368814). RNA was input equally (500ng/reaction) and samples were incubated according to the High Capacity cDNA Synthesis kit instructions to obtain cDNA.

Pre-designed FAM-labelled Taqman qPCR assays were purchased from ThermoFisher for methylenetetrahydrofolate reductase or *MTHFR* (Cat. No. 4331182, Assay Number Hs01114487_m1), DNA methyltransferase 3 alpha or *DNMT3A* (Cat. No. 4331182, Assay Number Hs01027162_m1), ten-eleven translocation methylcytosine dioxygenase 1 or *TET1* (Cat. No.4448892, Assay Number Hs04189344_g1), ten-eleven translocation methylcytosine dioxygenase 2 or *TET2* (Cat. No. 4448892, Assay Number Hs00325999_m1), ten-eleven translocation methylcytosine dioxygenase 3 or *TET3* (Cat. No. 4448892, Assay Number Hs00896441_m1), and methyl-CpG binding protein 2 or *MECP2* (Cat. No. 4351372, Assay Number Hs05049079_g1) and were used for qPCR. *MTHFR* was used to validate the efficacy of the siRNA treatment. *DNMT3A* was used to validate the array results, and *MECP2* was added since *MECP2* is widely involved in chromatin structure but was not on the array plate. The three *TET* genes' expressions were measured since they reverse DNA methylation and were not on the array plate. The endogenous control, beta actin or ACTB, was VIC labeled (ThermoFisher; Cat. No. 4326315E). The samples were loaded in equal amounts in quadruplicate. Taqman qPCR reactions were incubated at 50˚C for 10 minutes and 95˚C for 2 minutes followed by 40 cycles of 95˚C for 15 seconds and 60˚C for 1 minute. The ΔΔCt method was used to obtain relative abundance of each gene of interest.

For gene expression arrays, RT-PCR was performed using the RT2 First Strand Kit (Qiagen; Cat. No. 330401) to obtain cDNA. Instructions for cDNA synthesis were contained in the Qiagen handbook supplied with the RT-PCR kit. Epigenetic chromatin modifying enzyme array plates (Qiagen; Cat. No. PAHS-085Z) were used along with SYBR Green qPCR Mix (Qiagen; Cat. No. 330502). The qPCR program described in the Qiagen booklet was utilized for incubation of the array plates. The plates consist of primers for genes that encode chromatin modifying enzymes, including DNMTs (DNA methyltransferases), HATs (histone acetyltransferases), HMTs (histone methyltransferases), HDACs (histone deacetylases), histone phosphorylases, histone ubiquitinases, and DNA/histone demethylases. A list of genes with accession numbers is provided in Table 1.

Each array plate was loaded with cDNA from one sample. Twelve array plates were used in total for 12 samples (3 replicates per treatment group). The array plates were analyzed by hand; the housekeeping (control) gene was chosen by comparing Ct values across all array plates. *RPLP0* had the most similar Ct value, all near 20, for the 12 array plates. The genomic DNA contamination wells were checked for a Ct value of >35 to ensure there was no genomic DNA contamination. The ΔΔCt method was used to obtain relative abundance of each gene of

**Table 1. A list of genes examined for expression in the epigenetic modifying enzymes array plates from Qiagen.**

| Gene Names | Refseqs | Functional Group |
|---|---|---|
| KDM1A, KDM4A, KDM4C, KDM5B, KDM5C, KDM6B, MBD2 | NM_015013, NM_014663, NM_015061, NM_006618, NM_004187, NM_001080424, NM_003927 | Demethylase |
| DNMT1, DNMT3A, DNMT3B | NM_001379, NM_022552, NM_006892 | DNMT |
| ATF2, CDYL, CIITA, CSRP2BP, ESCO1, ESCO2, HAT1, KAT2A, KAT2B, KAT5, KAT6A, KAT6B, KAT7, KAT8, NCOA1, NCOA3, NCOA6 | NM_001880, NM_004824, NM_000246, NM_020536, NM_052911, NM_001017420, NM_003642, NM_021078, NM_003884, NM_006388, NM_006388, NM_006766, NM_012330, NM_007067, NM_032188, NM_003743, NM_181659, NM_014071 | HAT |
| HDAC1, HDAC2, HDAC3, HDAC4, HDAC5, HDAC6, HDAC7, HDAC8, HDAC9, HDAC10, HDAC11 | NM_004964, NM_001527, NM_003883, NM_006037, NM_005474, NM_006044, NM_001098416, NM_018486, NM_178425, NM_032019, NM_024827 | HDAC |
| CARM1, DOT1L, EHMT2, KMT2A, KMT2C, PRMT1, PRMT2, PRMT3, PRMT5, PRMT6, PRMT7, PRMT8, SETDB2, SMYD3, SUV39H1 | NM_199141, NM_032482, NM_006709, NM_005933, NM_170606, NM_001536, NM_001535, NM_005788, NM_006109, NM_018137, NM_019023, NM_019854, NM_031915, NM_022743, NM_003173 | HMT |
| ASH1L, KMT2E, NSD1, SETD1A, SETD1B, SETD2, SETD3, SETD4, SETD5, SETD6, SETD7, SETD8, SETDB1, SUV420H1, WHSC1 | NM_018489, NM_182931, NM_022455, NM_014712, NM_015048, NM_014159, NM_199123, NM_017438, NM_001080517, NM_024860, NM_030648, NM_020382, NM_012432, NM_016028, NM_007331 | HMT Activity (SET Domain Proteins) |
| AURKA, AURKB, AURKC, NEK6, PAK1, RPS6KA3, RPS6KA5 | NM_003600, NM_004217, NM_003160, NM_014397, NM_002576, NM_004586, NM_004755 | Phosphorylase |
| DZIP3, MYSM1, RNF2, RNF20, UBE2A, UBE2B, USP16, USP21, USP22 | NM_014648, NM_001085487, NM_007212, NM_019592, NM_003336, NM_003337, NM_006447, NM_012475, NM_015276 | Ubiquitinase |
| ACTB, B2M, GAPDH, HPRT1, RPLP0 | NM_001101, NM_004048, NM_002046, NM_000194, NM_001002, | Housekeeping Gene |
| HGDC | SA_00105 | Genomic DNA Control |

Accession numbers (Refseqs) and gene functional groupings are included.

interest. During statistical analyses, genes *AURKC*, *CIITA*, and *PRMT8* were not considered because their Ct values were >35 for all samples, indicating extremely low cDNA quantity in the SHSY5Y cell line.

## Nuclear extractions and enzymatic activity assays

Nuclear extracts were isolated using the Nuclear Extraction Kit (Epigentek; Cat. No. OP-0002-1). Three replicates were obtained per treatment group; each replicate contained the contents of twelve wells from two 6-well plates. Twelve samples were collected so that there were three replicates per treatment group. A Bradford assay was used to quantify total protein in each sample.

Enzyme activity assays from Epigentek for DNMTs (Cat. No. P-3009-48), HMTs (H3K4) (Cat. No. P-3002-1), HATs (Cat. No. P-4003-48), TETs (Cat. No. P-3086-48), and HDACs (Cat. No. P-4034-48) were used. Nuclear extracts were loaded in equal amounts into the plates in duplicate. Antibodies specific to each enzyme class were added and were incubated with the nuclear extracts. After washes, a secondary antibody with a conjugated reporter was added, followed by washing of the wells and addition of a substrate solution. A stop solution was added to inhibit further reporter-substrate interactions, and the wells were analyzed for their absorbance readings at the 450nm and 655nm wavelengths using a spectrophotometric plate reader. Data analyses were performed according to the handbook provided by Epigentek.

## Results

### DNA methylation & hydroxymethylation and histone 3 modifications

A two-way ANOVA with replication determined that the *MTHFR* KD had a significant effect on global DNA methylation (5-mC) compared to the SCR group (p = 0.0000042, Fig 1A). The

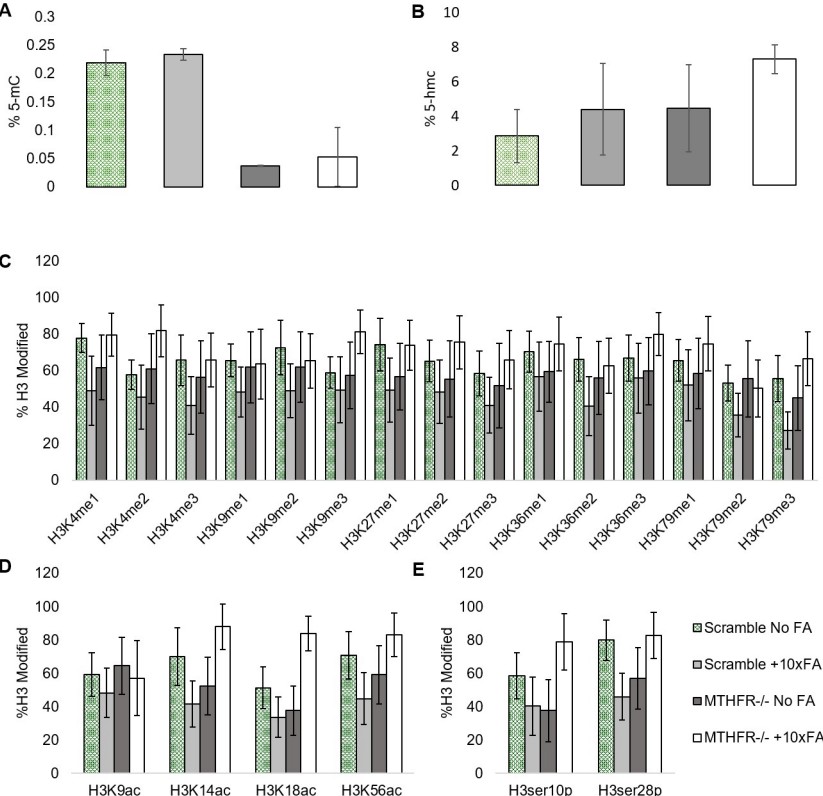

**Fig 1. DNA methylation and Histone modifications in the scramble + no FA treatment, the scramble +10x FA treatment, the *MTHFR* -/- +No FA treatment group (*MTHFR*-/- refers to the *MTHFR* knockdown), and the *MTHFR*-/-+10Xfa treatment group.** A. Percent of 5-methylCytosine detected globally in all treatment groups. B. Percent of 5-hydroxymethylCytosine detected globally in all treatment groups. C. Percent of Histone 3 methylation modifications examined via array in all treatment groups. D. Percent of Histone 3 acetylation modifications examined via array in all treatment groups. E. Percent of Histone 3 phosphorylation modifications examined via array in all treatment groups. N = 3 for each treatment group. * = p<0.05, ** = p<0.01. Error bars indicate +/- 1 Standard Deviation.

KD significantly decreased the 5-mC. The 10x FA treatment alone had no significant effect on 5-mC levels in the SHSY5Y cells (p = 0.38). The interaction did not significantly alter 5-mC levels (p = 0.99). The KD alone led to a significant increase in DNA hydroxymethylation (5-hmC) levels (p = 0.046; Fig 1B), while the FA alone did not affect 5-hmC levels significantly (p = 0.05). The interaction did not significantly affect 5-hmC levels (p = 0.53).

Histone 3 (H3) methylation analysis revealed no significant changes to any H3 methylation marks (Fig 1C). H3K18Ac marks were significantly increased due to the interaction between KD and FA (p = 0.005; Fig 1D). H3 serine (ser) analysis revealed significant changes in H3ser10p marks (Fig 1E). The KD alone induced a significant decrease in H3ser10p marks (p = 0.02, Fig 1E). The interaction of the KD and the FA treatment also caused a significant increase in H3ser10p marks (p<0.001, Fig 1E). There was also a significant increase in H3ser28p in the interaction between KD and FA treatment (p<0.001, Fig 1E).

## Gene expression analyses

Table 2 includes genes from the epigenetic chromatin modifying enzyme array plate that were significantly up or downregulated after a two-way ANOVA with replication. The FA treatment

**Table 2. P-values for each gene in the Qiagen gene expression array plate described in Table 1.**

| Gene Name | KD p-value | FA p-value | Interaction p-value |
|-----------|-----------|-----------|---------------------|
| ASHL1 | 0.207 | 0.165 | 0.573 |
| ATF2 | 0.859 | 0.893 | 0.443 |
| AURKA | 0.151 | 0.438 | 0.336 |
| AURKB | **0.005** | 0.575 | 0.059 |
| CARM1 | 0.626 | **0.018** | 0.291 |
| CDYL | 0.532 | **0.037** | 0.853 |
| CSRP2BP | **0.025** | 0.203 | 0.104 |
| DNMT1 | 0.054 | 0.436 | 0.153 |
| DNMT3A | 0.600 | **0.022** | 0.554 |
| DNMT3B | 0.312 | 0.062 | 0.496 |
| DOT1L | 0.182 | 0.080 | 0.737 |
| DZIP3 | 0.249 | 0.188 | 0.528 |
| EHMT2 | 0.485 | **0.028** | 0.782 |
| ESCO1 | 0.604 | **0.002** | 0.967 |
| ESCO2 | 0.429 | 0.128 | 0.256 |
| HAT1 | 0.135 | 0.206 | 0.141 |
| HDAC1 | 0.134 | 0.122 | 0.849 |
| HDAC10 | 0.260 | 0.130 | 0.389 |
| HDAC11 | 0.918 | **0.002** | 0.237 |
| HDAC2 | 0.173 | **0.004** | 0.501 |
| HDAC3 | 0.465 | 0.086 | 0.161 |
| HDAC4 | 0.132 | 0.066 | 0.994 |
| HDAC5 | **0.017** | **0.017** | 0.527 |
| HDAC6 | 0.377 | 0.182 | 0.091 |
| HDAC7 | 0.910 | 0.051 | 0.901 |
| HDAC8 | 0.101 | **0.027** | 0.159 |
| HDAC9 | 0.193 | 0.265 | 0.957 |
| KAT2A | 0.097 | 0.365 | 0.386 |
| KAT2B | 0.658 | **0.031** | 0.949 |
| KAT5 | 0.096 | 0.412 | 0.317 |
| KAT6A | 0.521 | 0.625 | 0.734 |
| KAT6B | 0.067 | 0.752 | 0.206 |
| KAT7 | 0.436 | **0.024** | 0.981 |
| KAT8 | 0.246 | 0.329 | **0.041** |
| KDM1A | 0.125 | 0.441 | 0.070 |
| KDM4A | 0.501 | 0.086 | 0.434 |
| KDM4C | 0.497 | **0.031** | 0.470 |
| KDM5B | 0.507 | **0.047** | 0.855 |
| KDM5C | 0.423 | **0.006** | 0.834 |
| KDM6B | **0.042** | **0.006** | 0.313 |
| MBD2 | 0.829 | **0.041** | 0.628 |
| KMT2A | 0.190 | 0.114 | 0.642 |
| KMT2C | 0.300 | 0.162 | 0.822 |
| KMT2E | 0.133 | 0.913 | 0.147 |
| MYSM1 | 0.194 | 0.068 | 0.223 |
| NCOA1 | 0.587 | 0.104 | 0.103 |
| NCOA3 | 0.137 | 0.240 | 0.384 |

(*Continued*)

**Table 2.** (Continued)

| Gene Name | KD p-value | FA p-value | Interaction p-value |
|-----------|-----------|-----------|---------------------|
| NCOA6 | 0.421 | 0.357 | 0.891 |
| NEK6 | 0.995 | 0.139 | 0.445 |
| NSD1 | 0.164 | **0.039** | 0.945 |
| PAK1 | 0.139 | 0.099 | 0.354 |
| PRMT1 | 0.574 | **0.046** | 0.837 |
| PRMT2 | 0.404 | **0.020** | 0.695 |
| PRMT3 | 0.359 | **0.008** | 0.240 |
| PRMT5 | 0.349 | 0.589 | 0.171 |
| PRMT6 | 0.168 | **0.022** | 0.823 |
| PRMT7 | **0.015** | 0.251 | 0.058 |
| RNF2 | 0.223 | 0.070 | 0.418 |
| RNF20 | 0.509 | **0.036** | 0.891 |
| RPS6KA3 | 0.613 | 0.261 | 0.447 |
| RPS6KA5 | 0.612 | 0.064 | 0.772 |
| SETD1A | 0.341 | **0.042** | 0.776 |
| SETD1B | **0.033** | **0.023** | 0.318 |
| SETD2 | 0.168 | 0.054 | 0.586 |
| SETD3 | 0.208 | 0.051 | 0.512 |
| SETD4 | 0.164 | 0.204 | 0.176 |
| SETD5 | 0.073 | 0.834 | 0.385 |
| SETD6 | **0.040** | 0.096 | 0.092 |
| SETD7 | 0.892 | 0.123 | 0.989 |
| SETD8 | 0.210 | **0.013** | 0.481 |
| SETDB1 | 0.455 | **0.017** | 0.521 |
| SETDB2 | 0.223 | 0.236 | 0.414 |
| SMYD3 | 0.080 | **0.011** | 0.317 |
| SUV39H1 | 0.179 | **0.014** | 0.656 |
| SUV420H1 | 0.065 | **0.017** | 0.451 |
| UBE2A | 0.123 | **0.013** | 0.123 |
| UBE2B | 0.242 | **0.009** | 0.699 |
| USP16 | 0.755 | **0.001** | 0.089 |
| USP21 | 0.191 | 0.157 | 0.530 |
| USP22 | 0.721 | 0.131 | 0.663 |
| WHSC1 | 0.527 | **0.035** | 0.788 |

P-values are given for the effect of the knockdown (KD) alone, 10x FA treatment alone (FA), and the interaction between the two independent variables. Bold font indicates a significant p-value.

induced a significant upregulation in many genes listed, with most significance existing upon FA treatment alone (general upregulation) or upon KD treatment alone (general downregulation). A clustergram generated by the data analysis center on the QIAGEN website (https://www.qiagen.com/us/shop/genes-and-pathways/data-analysis-center-overview-page/) is shown in Fig 2. Generally, the clustergram indicates increases in gene expression in the SCR +10x FA group compared to the SCR group and little overall change in the KD group, likely since there was an overall increase in gene expression in two of the samples but an overall decrease in the third sample.

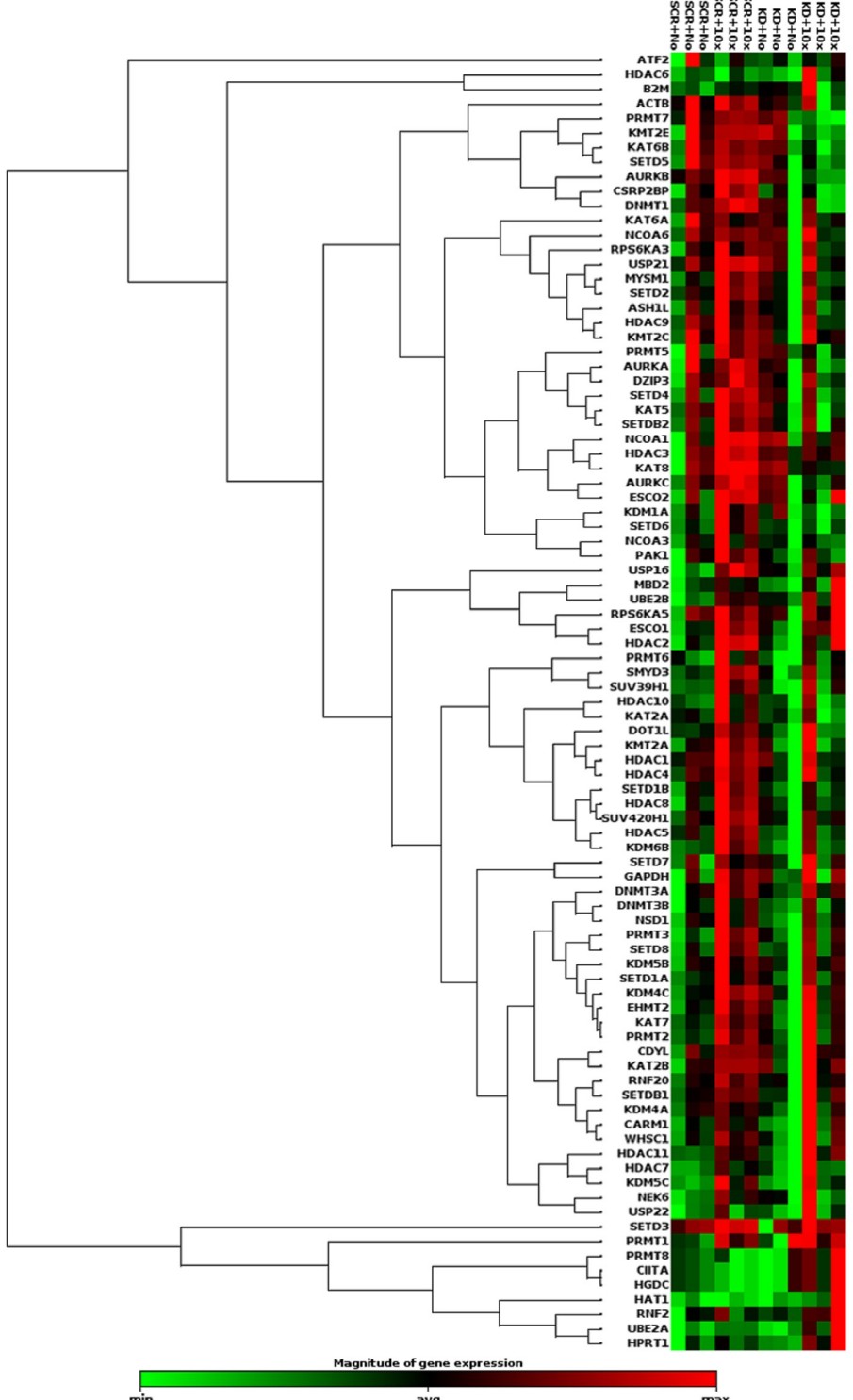

**Fig 2. Heat map generated by Qiagen's gene expression array software for epigenetic modifying enzymes.** As shown by the legend, red indicates an increase in gene expression while green indicates a reduction in gene expression. N = 3 for each treatment group.

## Taqman assays

The results of the *DNMT3A* Taqman assays align with the results for *DNMT3A* from the array plates. Following a two-way ANOVA with replication, it was evident that *DNMT3A* expression significantly decreased due to FA treatment alone (p = 0.03, Fig 3A). *MECP2* Taqman assays revealed *MECP2* expression significantly decreased due to FA treatment alone (p = 0.0000000389, Fig 3B) and increased due to KD treatment alone (p = 0.00011). Further, the interaction between the KD and excess FA treatment significantly reduced *MECP2* expression (p = 0.002). The *MTHFR* Taqman assay revealed that *MTHFR* expression significantly decreased due to the FA treatment alone (p = 0.000044, Fig 3C). *MTHFR* expression significantly decreased due to the KD alone as well, which indicates the knockdown was effective (p = 0.002). The interaction did not significantly affect *MTHFR* expression (p = 0.37). The *TET1* Taqman assays revealed no effects of the FA treatment on *TET1* expression (p = 0.62, Fig 3D) while the KD alone also had no effect (p = 0.82). The interaction had no effect on *TET1* expression (p = 0.38). *TET2* was not affected by the FA treatment alone (p = 0.80 Fig 3E) while the KD alone led to a significant increase in *TET2* (p = 0.01). The interaction of the KD and excess FA treatment did not affect *TET2* (p = 0.34). The KD alone did not affect *TET3* significantly (p = 0.21, Fig 3F) while the excess FA treatment alone decreased *TET3* expression (p = 0.002). The interaction between the KD and excess FA did not significantly affect *TET3* expression (p = 0.30).

## Dendritic spine density analysis

The KD alone significantly increased dendritic spine counts (p = 0.02, Fig 4A). The FA treatment alone did not significantly affect the spine counts (p = 0.21). Additionally, the interaction of the KD and the FA treatment significantly decreased the dendritic spine counts (p<0.001).

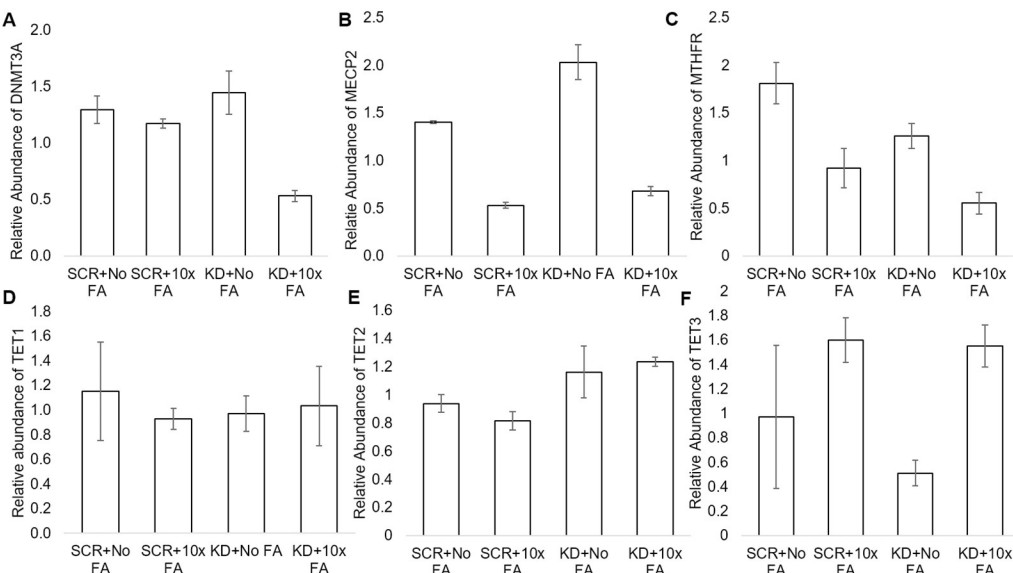

**Fig 3. Gene expression of genes tested via Taqman assays in SCR, SCR+10xFA, KD, and KD+10x FA treatment groups.** SCR refers to scramble + no FA treatment, SCR+10x FA refers to scramble +10x FA treatment, KD refers to *MTHFR* knockdown, and KD+10x FA refers to the *MTHFR* knockdown+10xFA treatment group. A. Relative abundance of *DNMT3A* in all test groups. B. Relative abundance of *MECP2* in all test groups. C. Relative abundance of *MTHFR* in all test groups. D. Relative abundance of *TET1* in all test groups. E. Relative abundance of *TET2* in all test groups. F. Relative abundance of *TET3* in all test groups. N = 3 for each treatment group. * = p<0.05, ** = p<0.01. Error bars indicate +/- 1 Standard Deviation.

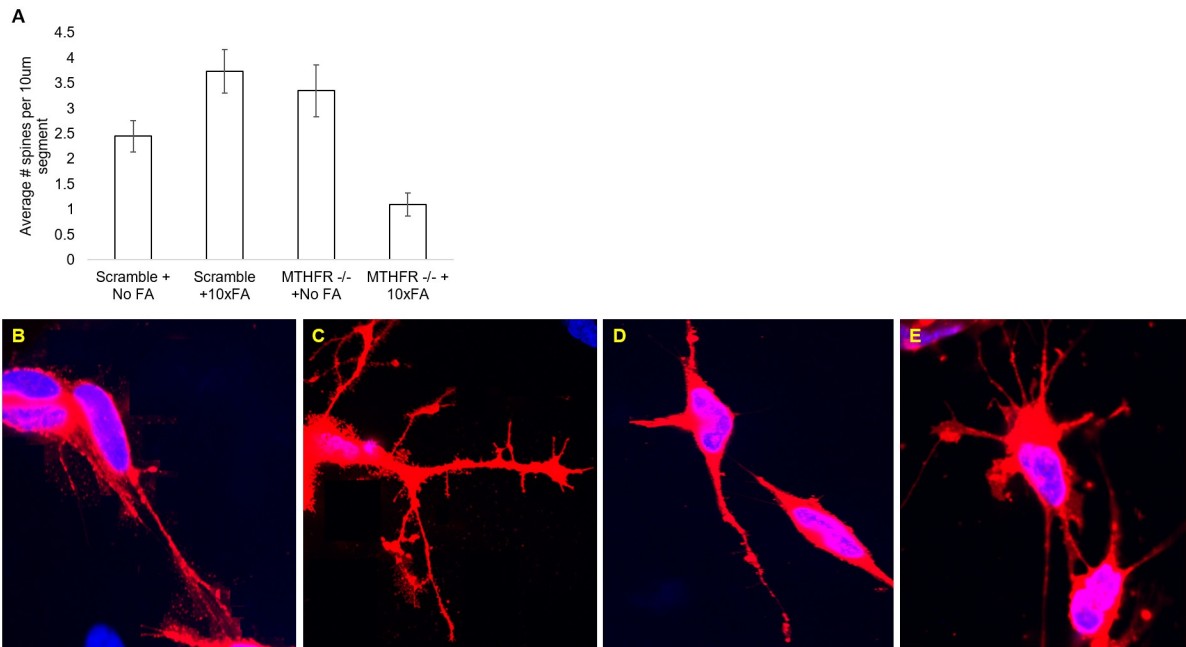

**Fig 4. Dendritic spine densities in the scramble + no FA treatment, scramble +10x FA treatment, *MTHFR* -/- + No FA treatment, and the *MTHFR* -/- +10x FA treatment group.** A. Dendritic spine densities (average) per 10um segment in each test group. B. Representative image of dendritic spines in the scramble + no FA group. C. Representative image of dendritic spines in the scramble + 10x FA group. D. Representative image of dendritic spines in the *MTHFR-/-* + no FA group. E. Representative image of dendritic spines in the *MTHFR-/-* + 10x FA group. N = 3 for each treatment group. * = p<0.05, ** = p<0.01. Error bars indicate +/- 1 Standard Deviation.

Representative images from confocal microscopy for each treatment group are in Fig 4B, 4C, 4D, and 4E.

## Enzymatic activity assays

DNMT activity significantly increased due to the KD alone (p = 0.004, Fig 5A). FA treatment did not significantly affect DNMT activity (p = 0.07), and the interaction between FA and KD was not significant (p = 0.44). The KD alone had no effect on H3K4 HMT activity (p = 0.96), and the same was true for FA treatment alone (p = 0.074). The interaction of KD and the FA treatment did not significantly affect H3K4 HMT activity (p = 0.36, Fig 5B). HDAC activity significantly decreased due to the interaction between the KD and FA treatment (p = 0.03; Fig 5C) though the KD (p = 0.3) and FA treatments (p = 0.8) alone had no significant effects. HAT activity was not significantly affected by the KD alone (p = 0.17, Fig 5D) or the FA treatment alone (p = 0.47). The interaction significantly decreased HAT activity (p = 0.042). The excess FA treatment alone had no effect on TET activity (p = 0.49, Fig 5E) as did the KD (p = 0.75). The interaction had no effect on TET activity as well (p = 0.67).

## Discussion

The purpose of this project was to determine if an *MTHFR* knockdown with or without a very high FA treatment (10x) resulted in changes in gene expression, H3 modifications, and/or dendritic spine densities in SHSY5Y cells. Additionally, the project aimed to determine the effects of the 10x FA treatment on SHSY5Y cells with normal *MTHFR* expression. The experiments presented here indicate the 10x FA treatment had various effects in both the normal (Scramble) and *MTHFR* knockdown groups. Additionally, the experiments presented here

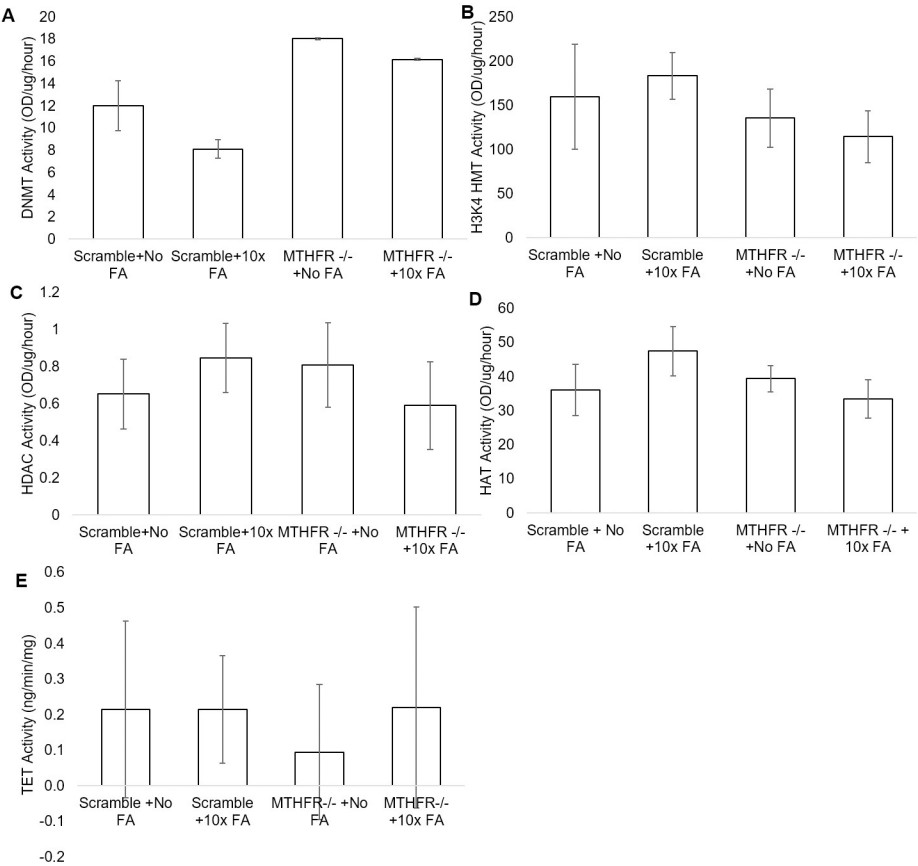

**Fig 5. Enzyme activity assays in the scramble + no FA treatment, scramble +10x FA treatment, MTHFR -/- + No FA treatment, and the MTHFR-/- +10x FA treatment groups.** A. DNMT activity in OD/ug/hour for all treatment groups. B. H3K4 HMT activity in OD/ug/hour for all treatment groups. C. HDAC activity in OD/ug/hour in all treatment groups. D. HAT activity in OD/ug/hour in all treatment groups. E. TET activity in ng/min/mg in all treatment groups. N = 3 for each treatment group. * = $p < 0.05$. Error bars indicate +/- 1 Standard Deviation.

indicate the effects of an *MTHFR* knockdown on H3 modifications, DNA methylation, and dendritic spines in SHSY5Y cells.

The gene expression array plates revealed that the 10x FA treatment alone had significant effects on gene expression since many genes were significantly increased due to FA treatment alone. This is evidenced by the clustergram in Fig 2 and data presented in Table 2. By name and function, the FA treatment alone increased expression of several DNA demethylases (*KDM4C*, *KDM5B*, *KDM5C*, *KDM6B*, and *MBD2*), a DNA methyltransferase (*DNMT3A*), several histone acetyltransferases (*CDYL*, *ESCO1*, *KAT2B*, and *KAT7*), several histone deacetylases (*HDAC2*, *HDAC5*, *HDAC8*, and *HDAC11*), several enzymes with histone methyltransferase-related activities (*CARM1*, *EHMT2*, *PRMT1*, *PRMT2*, *PRMT3*, *PRMT6*, *SUV420H1*, and *WHSC1*), and several ubiquitinases (*RNF20*, *UBE2A*, *UBE2B*, and *USP16*).

The KD alone had very little effect on genes' expression levels that were detected via the array plates. Only expression of *AURKB* (histone phosphorylase), *CSRP2BP* (histone acetyltransferase), *HDAC5* (histone deacetylase), *KDM6B* (DNA Demethylase), *PRMT7* (histone methyltransferase), *SETD1B* and *SETD6* (both histone methyltransferase activity enzymes) were significantly decreased by the KD alone. Interestingly, this list affected by the KD alone spans a wide variety of the types of genes tested in the array plate. The interaction between the

KD and the 10x FA treatment only significantly affected one gene's expression, *KAT8*, which is a histone acetyltransferase.

*MECP2* was reduced by the 10x FA treatment alone; there were reductions in *MECP2* expression in SCR+10x FA and KD+10x FA groups compared to the SCR group and the KD group, respectively. Decreased *MECP2* has been cited as a characteristic of prefrontal cortical neurons in autism spectrum disorder patients [23]. *MECP2* was significantly higher in the KD group when compared to the SCR group, indicating the *MTHFR* knockdown alone drives *MECP2* expression upwards. The interaction of the FA treatment and KD also was significant, as the interaction led to a decrease in *MECP2*. The higher than normal *MECP2* expression is reminiscent of overexpression of *MECP2* in *MECP2* duplication syndrome, which is known to be detrimental in the carrying offspring [24].

Dendritic spines changed significantly due to the KD alone and also due to the interaction between the KD and the 10x FA treatment. An increase in dendritic spine densities when comparing the SCR group to SCR+10x FA group makes sense in light of the decrease in *MECP2* expression in the SCR+10x FA group since decreased *MECP2* is associated with a decrease in spine densities in neurons [25, 26]. The increase in spines in the SCR+10x FA group compared to the SCR group also aligns with our previous findings using a 2x FA treatment [21]. The confounding evidence was the KD+10x FA had a lower average spine count compared to SCR and KD, which does not fit the model with decreased *MECP2*. Evidently, another factor must have been involved in the decrease in spine density in the KD+10x FA group.

The *MTHFR* Taqman assay results indicated (1) the siRNA knockdown was effective, and (2) FA caused a decrease in *MTHFR* expression, as indicated by the reduction in *MTHFR* due to the KD treatment alone and the 10x FA treatment alone, respectively. This matches previous findings in other research that indicate high FA decreases *MTHFR* further [15].

The *MTHFR* KD alone decreased 5-mC, but the 10x FA appears to have no effect on 5-mC. This is interesting considering the *DNMT* gene expression levels were significantly higher with 10x FA treatment and because enzyme activity was only significantly affected by the KD alone. This indicates there may be an effect on mRNA degradation, possibly by microRNAs (miRNAs), which were not studied here, or an effect on protein degradation via ubiquitination and proteases. It is notable that several genes whose products assist with protein degradation involving ubiquitination were significantly upregulated due to the FA treatment alone. Further, since *MECP2* has been shown to have various effects on methylation activity and gene expression, changes to *MECP2* expression could also have affected 5-mC levels too [25, 27].

DNA hydroxymethylation (5-hmC) was not significantly affected by the FA alone as our alpha is set to $p < 0.05$, and for the FA alone, $p = 0.05$. 5-hmC was significantly increased due to the KD alone, and interestingly, *TET2* expression significantly increased due to the KD alone. The FA alone led to a significant decrease in *TET3* expression. *TET1* expression was unaffected in our assays. The TET enzyme activity assays revealed no significant changes to TET enzymes overall, which does not quite align with the gene expression assay data. The overall TET enzyme activity assay data does not align with the *TET* mRNA expression data or the 5-hmC data. TET enzyme activity was at a very low level according to our assay. Therefore, it is possible the assay is not as sensitive, and malalignment of the TET activity assay results with the gene expression data is due to overall measurement of TET enzymes (rather than measuring individual enzymes' activity). It is still noteworthy that 5-hmC increased due to the KD (while the KD alone decreased 5-mC). 5-hmC levels may have affected DNA methylation (5-mC) levels as well, since the TET enzymes play a role in the reversal of DNA methylation via hydroxylation of methylated DNA.

There was no significant effect on H3 histone methylation. This finding aligns with the lack of significant change to H3K4 HMT activity, though the assay was specific for only H3K4

HMTs. H3K18Ac (histone 3, lysine 18 acetylation) increased significantly due to the interaction between the KD and the 10x FA treatment. It is notable there is an increase in the KD +10x FA group compared to SCR+10x FA group. The high FA exposure affected H3K18Ac only when the cells had a knockdown of *MTHFR*, since the KD+10x FA group had significantly more H3K18Ac compared to the KD group. The acetylation data makes sense in light of the significant decrease in HDAC activity due to the interaction, though there was decreased HAT activity due to the interaction between the KD and the 10x FA treatment. Interestingly, high H3K18Ac is associated with cellular senescence, cancer, and poor outcomes in cancers such as hepatocarcinoma and pancreatic cancer [28–31]. However, higher H3K18Ac in the KD+10x FA group compared to the KD group may hint towards the protective effect of high FA against certain neural tube defects (NTDs) since amniotic fluid and sera of NTD pregnancies had low H3K18Ac levels [32].

The KD alone induced a significant decrease in H3ser10p (histone 3, serine 10 phosphorylation) marks, but the interaction between the KD and FA treatment caused a significant increase in H3ser10p marks and H3ser28p marks. Increases in H3ser10p promote cell cycle progression at the G2/M phase transition [33]. As autism spectrum disorder patients have increased neuron counts in the prefrontal cortex [23] and higher rates of *MTHFR* mutations [34], our results suggest too much FA when *MTHFR* is functionally decreased may have detrimental effects on neurological development.

Upon studying all of the data gathered, it was evident gene expression at the mRNA level was not indicative of protein levels or function as much of the gene expression data does not align with the DNA and/or histone modification data or enzyme activity assays. We hypothesize the FA treatment and/or *MTHFR* knockdown may affect posttranscriptional regulation and/or posttranslational modifications (e.g. miRNAs and ubiquitination, respectively) to regulate enzyme functions, histone modifications, and DNA methylation levels. This would make sense in light of our findings that Ubiquitinases were significantly increased with the 10x FA treatment. Further, the genes on the gene expression array plates from Qiagen may not match the enzymes that were tested in enzyme activity assay kits from Epigentek, and the kits analyzed many enzymes at once.

Overall, the *MTHFR* knockdown had significant effects on *MECP2* and *TET2* mRNA expression levels, global 5-mC levels, global 5-hmC levels, H3K18Ac levels, and H3ser10p levels. The KD alone also led to an increase in dendritic spines. The knockdown appeared to have some combinatory effects with the high FA treatment on dendritic spines, which decreased, *MECP2* expression, HDAC activity, and HAT activity. The high FA treatment alone affected the gene expression of many chromatin modifying enzymes, as well as *MECP2*, *TET3*, and *MTHFR* expression at the mRNA level. A high FA exposure also increased dendritic spine density, though not significantly, and high FA affected H3K18Ac levels and H3ser10p levels. Of note, increases in H3K18Ac and H3ser10p levels are harmful in some instances, but might be helpful in other instances, and this likely depends on cell type. Future studies should examine whether rescuing the expression of the MTHFR gene would reverse the epigenetic changes witnessed in this study. Further, analysis of epigenetic marks, cell survival, and cell differentiation in stem cells would be an interesting avenue for further research in regards to the KD and the excess FA treatments.

## Conclusions

In conclusion, it is suggested that high levels of FA in the presence or absence of *MTHFR* function might have negative effects in cells, especially mitotic cells, of a developing organism. This is especially true as FA consumption is much higher in more recent years, and many people

are unaware of a mutation in *MTHFR*, especially if they are heterozygous. There is the potential that this study has meaning for human health. Our results indicate that further research is needed to determine if over-supplementation with FA is truly detrimental at the organismal level, with special care to note presence or absence of *MTHFR* mutations. There is much still to learn about how the KD and/or high FA would affect stem cell epigenetic marks, cell differentiation, and cell survival, and how this relates to the development of an organism *in utero*.

## Supporting information

**S1 File. This file contains all data presented in this paper.**
(XLSX)

## Acknowledgments

We would like to thank Thomas Davis at Gibbs Cancer Center for assistance with confocal microscopy imaging. We would like to thank Gabriel Castillo and Hunter McLeod for assistance during some of the ELISA-based enzyme activity assays. We would also like to thank Dr. Jeannie Chapman, Dean of the College of Science and Technology at USC Upstate, for supporting this work while still chair of the Division of Natural Sciences and Engineering at USC Upstate. We finally would like to thank Dr. Ben Montgomery, current chair of the Division of Natural Sciences and Engineering, for support of this work.

## Author Contributions

**Conceptualization:** Kimberly R. Shorter.

**Data curation:** Kimberly R. Shorter.

**Formal analysis:** Daniel F. Clark, Rachael Schmelz, Nicole Rogers, Nuri E. Smith, Kimberly R. Shorter.

**Funding acquisition:** Daniel F. Clark, Rachael Schmelz, Kimberly R. Shorter.

**Investigation:** Daniel F. Clark, Rachael Schmelz, Nicole Rogers, Nuri E. Smith, Kimberly R. Shorter.

**Methodology:** Kimberly R. Shorter.

**Project administration:** Kimberly R. Shorter.

**Supervision:** Kimberly R. Shorter.

**Writing – original draft:** Kimberly R. Shorter.

**Writing – review & editing:** Kimberly R. Shorter.

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
