## [Decision Letter · Decision Letter 0]

28 Aug 2020

PONE-D-20-23387

Acute high folic acid treatment in SH-SY5Y cells with and without MTHFR function leads to gene expression changes in epigenetic modifying enzymes, changes in epigenetic marks, and changes in dendritic spine densities.

PLOS ONE

Dear Dr. Shorter,

Thank you for submitting your manuscript to PLOS ONE. After careful consideration, we feel that it has merit but does not fully meet PLOS ONE’s publication criteria as it currently stands. Therefore, we invite you to submit a revised version of the manuscript that addresses the points raised during the review process.

Please respond to all Reviewer's comments paying particular attention to Reviewer's comment n.2

We look forward to receiving your revised manuscript.

Kind regards,

Lorenzo Chiariotti

Academic Editor

PLOS ONE

Journal Requirements:

2. Quality of Figures must be improved

Additional Editor Comments (if provided):

Reviewers' comments:

Reviewer's Responses to Questions

**Comments to the Author**

1. Is the manuscript technically sound, and do the data support the conclusions?

Reviewer #1: Yes

2. Has the statistical analysis been performed appropriately and rigorously? 

Reviewer #1: Yes

3. Have the authors made all data underlying the findings in their manuscript fully available?

Reviewer #1: Yes

4. Is the manuscript presented in an intelligible fashion and written in standard English?

Reviewer #1: Yes

5. Review Comments to the Author

Reviewer #1: In this work, the authors investigated gene expression and activity of epigenetic modifying enzymes, genome-wide DNA methylation, histone 3 modifications, and dendritic spine densities in SH-SY5Y cells with or without a knockdown of MTHFR and with or without an excess of folic acid. The paper is well written and the experiments performed by the authors are distinctly explained. The discussion of the results is fluent so that the reader can easily understand the goals of the authors. Very few notes are required:

1) The authors provided an epigenetic characterization of SH-SY5Y cells knocked down for MTHFR. Did the authors evaluated whether the rescue of the KD maintain this epigenetic changes, especially for MecP2 protein?

2) The authors described an increased expression of several DNA demethylases that well match with the observed decrease in global DNA methylation. In order to complete the scenario, the authors could also measure the level of hydroxymethyl-C and the expression levels of TET’s family enzymes both in KD and control cells after treatment with FA.

3) It would be very interesting to understand what happens in stem cells treated with FA and Knocked Down for MTHFR. Could they differentiate?

4) Please, adjust the resolution of all the figures.

6. PLOS authors have the option to publish the peer review history of their article (what does this mean?). If published, this will include your full peer review and any attached files.

Reviewer #1: No

---

## [Author Response · Author response to Decision Letter 0]

10 Nov 2020

Addressing editorial comments:

1) We have updated the formatting of our manuscript document to reflect the formatting requests of the journal. 

2) We have formatted images to be better quality, and the images now meet the requirements for figures set forth by the journal. 

Addressing reviewer comments: 

1) We agree that evaluating the effects of rescuing the MTHFR knockdown would be an interesting investigation. Unfortunately, we do not have the funding/resources to do these experiments at the moment, but we have addressed the reviewer’s comment by suggesting this as a future study in the discussion and conclusions section. 

2) We agree that evaluating DNA hydroxymethylation and expression of TET enzymes would benefit this paper, and we did have the resources readily available to carry out such experiments. Most of the changes in our manuscript are due to adding (1) a study on DNA hydroxymethylation levels (global 5-hmC), (2) a study on the gene expression levels of TET1, TET2, and TET3 using Taqman assays, and (3) a study on enzyme activity for TETs. 

3) We agree that evaluating the effects of the knockdown with and without the excess folic acid on a stem cell population would be an extremely interesting study. Unfortunately, repeating our experiment in a stem cell population was not an option at the moment due to financial constraints. We did address the reviewer comment by suggesting a study on stem cell differentiation, survival, and epigenetic marks as a potential future direction in our discussion and conclusions section. 

4) We reformatted our images to be higher quality images with properties that align with the journal’s instructions.

---

## [Decision Letter · Decision Letter 1]

21 Dec 2020

Acute high folic acid treatment in SH-SY5Y cells with and without MTHFR function leads to gene expression changes in epigenetic modifying enzymes, changes in epigenetic marks, and changes in dendritic spine densities.

PONE-D-20-23387R1

Dear Dr. Shorter,

We’re pleased to inform you that your manuscript has been judged scientifically suitable for publication and will be formally accepted for publication once it meets all outstanding technical requirements.

Kind regards,

Lorenzo Chiariotti

Academic Editor

PLOS ONE

Additional Editor Comments (optional):

Reviewers' comments:

Reviewer's Responses to Questions

**Comments to the Author**

1. If the authors have adequately addressed your comments raised in a previous round of review and you feel that this manuscript is now acceptable for publication, you may indicate that here to bypass the “Comments to the Author” section, enter your conflict of interest statement in the “Confidential to Editor” section, and submit your "Accept" recommendation.

Reviewer #1: All comments have been addressed

2. Is the manuscript technically sound, and do the data support the conclusions?

Reviewer #1: Yes

3. Has the statistical analysis been performed appropriately and rigorously? 

Reviewer #1: Yes

4. Have the authors made all data underlying the findings in their manuscript fully available?

Reviewer #1: Yes

5. Is the manuscript presented in an intelligible fashion and written in standard English?

Reviewer #1: Yes

6. Review Comments to the Author

Reviewer #1: (No Response)

7. PLOS authors have the option to publish the peer review history of their article (what does this mean?). If published, this will include your full peer review and any attached files.

Reviewer #1: **Yes: **Mariella Cuomo

---

## [Editor Report · Acceptance letter]

28 Dec 2020

PONE-D-20-23387R1 

Acute high folic acid treatment in SH-SY5Y cells with and without *MTHFR* function leads to gene expression changes in epigenetic modifying enzymes, changes in epigenetic marks, and changes in dendritic spine densities. 

Dear Dr. Shorter:

I'm pleased to inform you that your manuscript has been deemed suitable for publication in PLOS ONE. Congratulations! Your manuscript is now with our production department. 

Kind regards, 

on behalf of

Dr. Lorenzo Chiariotti 

Academic Editor

PLOS ONE